# Socioeconomic inequalities in overweight and obesity among adults: Results from the Indonesian Food Barometer study

Siti Nurokhmah[1]*, Judhiastuty Februhartanty[2,3], Helda Khusun[3,4]

1 Department of Nutrition Science, Faculty of Health Science, Universitas Muhammadiyah Surakarta, Surakarta, Indonesia, 2 Department of Nutrition, Faculty of Medicine, Universitas Indonesia—Dr. Cipto Mangunkusumo General Hospital, Jakarta, Indonesia, 3 SEAMEO Regional Center for Food and Nutrition (RECFON)/Pusat Kajian Gizi Regional (PKGR) Universitas Indonesia, Jakarta, Indonesia, 4 Faculty of Health Sciences, University of Muhammadiyah Prof. Dr. HAMKA, Jakarta, Indonesia

* siti.nurokhmah@ums.ac.id

## Abstract

### Background

Globally, overweight and obesity (OAO) are significant health concerns that have detrimental effects throughout an individual's lifetime. This study examined socioeconomic inequality in OAO among Indonesian adults.

### Methods

Data were extracted from 1602 Indonesian adults who participated in the 2018 Indonesian Food Barometer study conducted in six provinces of Indonesia. Participants with a body mass index of 23 kg/m² or higher were classified as OAO. Socioeconomic-related inequality in OAO was estimated using the Wagstaff normalised concentration index (WCI) using wealth status and education level as the equity stratifiers. The index was further decomposed to find factors explaining the inequality.

### Results

Of the total participants, 56.3% (95% confidence interval [CI] 52.3% to 60.2%) lived with OAO. There were more of them among poor participants (WCI −0.073, CI 95% −0.129 to −0.017) and the more educated (WCI 0.062, 95% CI 0.008–0.117). The inequality by wealth status and education level only occurred among males with WCI −0.105 (95% CI −0.207 to −0.003) and 0.224 (95% CI 0.129 to 0.306), respectively. The main contributors to these wealth-related inequalities were wealth status (79.6%), residence (81.4%) and education (29.7%), while sex and province seemed to reduce the observed inequality, with negative contributions of −20.4% and −18.7%, respectively. By education level, the observed inequality was mostly explained by

**Data availability statement:** All relevant summary data necessary to reproduce our findings are provided within this paper and its Supporting Information files (see S1 Table). Access to the raw individual-level data is restricted under the Regulation of the Minister of Health of the Republic of Indonesia No. 85 of 2020 on the Transfer and Use of Materials, Information Content, and Data, to which SEAMEO RECFON complies. Interested researchers may request access to the raw data by contacting SEAMEO RECFON at research@seameo-recfon.org. Access will be granted only upon institutional approval and the completion of a formal Data Use Agreement.

**Funding:** The author(s) received no specific funding for this work.

**Competing interests:** The authors have no conflicts of interest associated with the material presented in this paper.

education (183.1%), residence (99.4%), age (−68.6%), marital status (−45.1%), religion (42.5%), and sex (−32.1%).

## Conclusions

Overall, we found that OAO is concentrated among the poor, but more educated population. Policies need to be developed to reduce the current burden of, and to close the socioeconomic gaps in OAO, especially among the underprivileged population and women.

## Introduction

Obesity is a chronic, complex, and preventable disease, and it has been an increasingly major public health concern worldwide, with prevalence continuing to rise in almost all regions [1]. In 2022, around 890 million people, which accounts for 16% of the global adult population, lived with obesity. Additionally, overweight is prevalent among 43% of the global population [2]. Along with their substantial adverse effects on health, obesity and overweight (OAO) are a global problem due to their significant financial burden on health systems, particularly in low- and middle-income countries (LMICs), as well as their social implications, which include stigma and social isolation [3].

Research on the association between socioeconomic status and OAO indicates a complex relationship, in which the prevalence and distribution of OAO vary across different regions, populations, and socioeconomic indicators. In high-income countries (HIC), OAO affects both men and women, all age categories, and has more significant influences on the less affluent groups [4,5]. In HICs, people living with OAO are more common among lower-income groups because limited access to healthy foods leads to higher consumption of calorie-dense, processed foods that are more affordable [6]. Moreover, these groups experience higher levels of stress due to socioeconomic pressure that can lead to emotional eating and weight gain [7]. As countries transition economically, the burden of obesity can shift from higher SES groups to lower SES groups, and we can see that the shift is often seen as countries move from low to middle-income status [5,8]. Data show that in middle-income countries, the relationship between SES and OAO becomes more complex: for men, the association is often mixed, while for women, it tends to be negative like the general population in HICs [8,9].

The rapid economic development, together with urbanisation and food system transformation, result in a phenomenon called nutrition transition—a shift in food consumption which promotes OAO and eventually the prevalence of chronic non-communicable diseases (NCDs) [4]. Indonesia national surveys reveal that the people's consumption pattern indicates the sign of nutrition transition: more westernized diets with higher saturated fats from animal sources, refined carbohydrates, added sugar content, and lower intake of whole grains. Consequently,

the country faces the double burden of malnutrition where undernutrition remains a public health concern but the OAO prevalence increases [10].

The percentage of Indonesian adults classified as OAO was 35.5% in 2018, which is a notable increase from 28.9% in 2013 [11]. This indicates a doubling over the course of the previous two decades [11]. Out of the affected population, adult women are disproportionately impacted. In 2018, 44.4% of women were living with OAO, while only 26.6% of men were affected. The findings also indicate an increase in the occurrence of OAO across all groups, even those who were historically less prone to OAO, such as low-income households and individuals residing in rural areas. Furthermore, this trend is observed in all provinces, including those with a significant prevalence of stunting and wasting, exacerbating the situation [11].

Several studies analyse the prevalence of and the role of socioeconomic factors on OAO in Indonesia indicating a remarkable increase across all population groups, particularly among women [12–16]. These studies show some evidence on the socioeconomic inequality in OAO, such as that OAO are more common in urban areas, wealthier, and highly educated populations. This socioeconomic inequality also existed among populations in Bangladesh, Cambodia, India, Myanmar, Nepal, Pakistan, and Timor-Leste, according to a previous analysis [17]. Nevertheless, there have been just a few studies that have made an effort to quantify the socioeconomic inequality in OAO in Indonesia [15,16]. Accordingly, the primary objective of this study was to evaluate socioeconomic inequalities in OAO and determine the key factors that contribute to the observed inequalities.

## Methods

### Data source and sampling

The 2018 Indonesian Food Barometer (IFB) survey collected data from December 2017 to March 2018 in six provinces (West Sumatra, Jakarta, West Java, East Java, Bali, and South Sulawesi), which together account for 48% of the Indonesian population [18]. The survey used a multistage sampling procedure, in which the first stage was selecting 6 out of 14 most populated provinces representing the most metropolitan area, ethnic diversity, and the eastern area of the country. Secondly, one rural and urban district were randomly selected in each selected province followed by the selection of clusters (villages) based on probability proportional to size. After that, random sampling was applied to select 4–5 hamlets in each village, in which households were then randomly selected from these hamlets. An eligible respondent for each household was randomly selected among all household members aged ≥18 years. We excluded pregnant women and respondents with missing information, yielding a sample of 1,604 cases (see S1 Table for detailed sample characteristics and data completeness). The IFB report provides detailed information on the sampling procedure and study design [18].

### Outcome variable

The outcome was a binary variable that indicated whether the person was overweight (with a Body Mass Index (BMI) between 23 kg/m$^2$ and less than 27.5 kg/m$^2$) or obesity (with a BMI of 27.5 kg/m2 or higher). We employed the cut-off values suggested by the World Health Organization (WHO) for the majority of the Asian population [19].

### Socioeconomic status

We employed two variables—wealth status and education level—as the criteria for categorizing equity or socioeconomic status (SES) in this analysis. The wealth variable was derived from household assets through the application of principal component analysis. The assets encompassed possession of electronic devices, livestock, and various amenities, such as a water source, cooking fuel, and materials for the floor, walls, and roof. The score was determined by utilising the factor score derived from principal component analysis (PCA) with varimax rotation [18]. Subsequently, we categorized these scores into quintiles, ranging from Q1 (representing the 20% of individuals with the lowest scores) to Q5 (representing

the 20% of individuals with the highest values). The education level was categorized into three groups: primary education or below (primary school or no formal education), secondary education (lower and upper secondary school), and higher education (college or university).

## Independent variables

We included several socio-demographic characteristics as the covariates: sex, age (18–25, 26–35, 36–45, ≥ 46 years), occupation (Professional, White-collar, Blue-collar, Not working [includes housewives and students]), marital status (single, married, divorced/widowed), having a child (yes, no), religion (Muslim, others), ethnicity (Minangkabau, Betawi, Sundanese, Javanese, Balinese, Bugis/Makassar, others), place of residence (urban, rural), and province (West Sumatra, Jakarta, West Java, East Java, Balinese and South Sulawesi). Wealth status and education level were also fitted as covariates.

## Statistical analysis

After data cleaning, we performed descriptive analysis to obtain the distribution of the sample and the prevalence of OAO by characteristics. Prior to the inequality analysis, we performed logistic regression to identify factors significantly associated with OAO. The results informed the selection of covariates included in the decomposition of the concentration index. The multicollinearity was evaluated using the variance inflation factor, and results showed no indication of it. We employed the concentration index (CI) to measure socioeconomic inequality in OAO. The CI values range from −1–1, where −1 and 1 represent the highest amount of disparity, while 0 signifies the lack of inequality. The positive CI indicates that the outcome is primarily observed in the most privileged group, whereas a negative CI suggests that the outcome is disproportionately observed in the disadvantaged groups of the rank variable. The measure of inequality is precisely defined by Equation 1.

$$CI = \frac{2}{\mu} \, cov(y, \, r)$$

(1)

The variable y represents the outcome variable, μ represents the mean or proportion of the outcome, r is the individual's fractional rank in the rank variable distribution, and cov refers to the covariates [20]. The CI was standardised by dividing it by $(1 - \mu)$ using the methodology employed by Wagstaff et al. for binary outcomes. This is because the CI value does not range from −1–1 but rather equal to $\mu - 1$ and $1 - \mu$ [21]. We employed the 'conindex' command to obtain the Wagstaff normalized concentration index (WCI) [22]. The WCI is one the most recommended methods for evaluating binary health outcomes in many studies in health economics and health services as it addresses the limitation of standard concentration index [23]. Lastly, the WCIs were further decomposed to evaluate the factors that contribute to the inequalities, using Equation 2.

$$WCI = \sum_k \left( \frac{\frac{\beta_k \bar{X}_k}{\mu}}{1-\mu} \right) C_k + \frac{\frac{GC_\varepsilon}{\mu}}{1-\mu} \; = \; \sum_k \left( \frac{\eta_k}{1-\mu} \right) C_k + \frac{\frac{GC_\varepsilon}{\mu}}{1-\mu}$$

(2)

The decomposition analysis yielded two components: the residual or unexplained component ($GC_\varepsilon$) and the explained component. The latter is derived by multiplying the elasticity ($\eta_k$) of each covariate ($X_k$) by the corresponding concentration index or WCI ($C_k$). Negative elasticities suggest that the outcome would be lower in that particular category when compared to the reference category. The WCI contribution was calculated by multiplying the elasticity with the WCI. The percentage of WCI was then determined by dividing the contribution by the total WCI. The unexplained component refers to the remaining portion of the WCI that cannot be accounted for by the covariates included in the decomposition model.

All analyses were conducted using Stata version 15 (StataCorp, College Station, Texas) with a statistical significance level of $p < 0.05$. We used Stata's 'svy' command to adjust for the study design by including sample weights, strata, and clusters.

### Ethical considerations

This study was approved by the Ethics Committee of the Faculty of Medicine, Universitas Indonesia (reference number 927/UN2. F1/ETIK/2017). A written informed consent was obtained from each participant prior to the interview.

### Results

Table 1 shows the distribution of study participants, the prevalence of OAO, the crude odds ratio (COR) and adjusted odds ratio (AOR) of OAO. The prevalence of OAO was 56.3% (95% CI 52.3% to 60.2%). Among the respondents, females were slightly higher than their counterparts, at 47.0% and 53.0%, respectively, while by age groups, the study participants were almost evenly distributed across categories. Females had a significantly higher prevalence of OAO (69.2%) and nearly threefold greater odds of OAO than males (AOR 2.77; 95% CI 1.93–3.98). Respondents with higher education represented the smallest group (11.2%), yet 63.3% of them were classified as living with OAO. Additionally, our analysis demonstrated positive associations of education and age with OAO. According to the wealth status, the study participants were almost equally distributed within the five categories. Further, although the percentage of OAO among the poorest or Q1 group (66.6%) was higher than that of the wealthiest group (55.5%), the adjusted analysis showed that the difference was not statistically significant (AOR 0.71(0.46–1.08). However, there was some evidence that respondents in the Q2–Q4 categories were less likely to be living with OAO compared to those in the Q1 group. The urban population accounted for approximately two-thirds of the sample and had higher odds of living with OAO (AOR 1.50; 95% CI 1.02–2.21). The percentage of respondents from East and West Java were much higher than the rest provinces; and the prevalence of OAO did not differ by this variable.

Ranked by wealth status and education level, the WCI of OAO for the overall sample were −0.073 and 0.062, respectively (Table 2). The negative WCI indicates that OAO was more concentrated among participants from more disadvantaged groups, i.e., poorer households or lower education levels, and vice versa. Our subgroup analysis also suggested that the disproportionate distribution of OAO across wealth status and education levels were only significant among males.

Contribution of the factors to the inequalities in OAO is summarised in Table 3. Our results suggested that Muslims, the richest 80% of participants, and those from West Sumatera, West Java, Bali, and South Sulawesi, had a lower probability of living with OAO compared to the reference group, as indicated by negative elasticity values. Conversely, single and married respondents, urban residents, and participants from East Java were more likely to be living with OAO. A negative WCI indicates the outcome is more concentration among the lower level of rank variable and vice versa. For example, in urban areas, OAO were more concentrated among less affluent households (WCI −0.279) and more educated population (WCI 0.280). The results also suggested that the largest contributors of the wealth-related inequalities in OAO were place of residence (81.4%), wealth status (79.8%), education level (29.7%), and sex (−20.4%). For the education-related inequality, education level and place of residence also had the highest contribution, while age and marital status played the biggest role in reducing the inequality as indicated by the negative contribution of −68.6% and −45.1%, respectively.

### Discussion

This study aimed to examine socioeconomic inequality in OAO among Indonesian adults. The results indicated that OAO were more concentrated among the poorer and more educated population. The major contributing factors of the observed inequalities were education level, place of residence, wealth status, and sex.

Our result of pro-poor wealth-related inequality in OAO was in line with a previous study [24]. Analysis on the Indonesian population using data from 1993 to 2014 highlights that the magnitude of pro-rich inequality in OAO has become

**Table 1. Prevalence of and factors associated with overweight and obesity (N = 1602).**

| Variables | n (%) | OAO % (95% CI) | COR (95% CI) | AOR (95% CI) |
|---|---|---|---|---|
| Sex | | | | |
| Male | 844 (53.0) | 44.8 (38.6-51.1) | Ref. | Ref. |
| Female | 758 (47.0) | 69.2 (64.1-73.9) | 2.77***(1.96-3.92) | 2.77***(1.93-3.98) |
| Age group | | | | |
| 18-25 | 338 (21.4) | 39.4 (31.0-48.6) | Ref. | Ref. |
| 26-35 | 446 (27.7) | 56.0 (47.5-64.2) | 1.96*(1.12-3.43) | 1.75*(1.07-2.84) |
| 36-45 | 319 (20.7) | 62.4 (53.3-70.7) | 2.55***(1.62-4.02) | 2.36***(1.53-3.65) |
| ≥ 46 | 499 (30.2) | 64.3 (58.8-69.4) | 2.76***(1.81-4.21) | 2.54***(1.66-3.88) |
| Education level | | | | |
| Primary or lower | 662 (45.1) | 53.6 (47.6-59.5) | Ref. | Ref. |
| Secondary | 699 (43.7) | 57.3 (52.9-61.6) | 1.16(0.89-1.51) | 1.55**(1.15-2.07) |
| Higher | 241 (11.2) | 63.2 (55.1-70.6) | 1.48*(1.06-2.07) | 1.77*(1.05-3.00) |
| Occupation | | | | |
| Professional | 54 (2.6) | 49.0 (29.9-68.3) | 0.63(0.28-1.41) | |
| White-collar | 507 (30.6) | 60.1 (51.9-67.7) | 0.99(0.72-1.36) | |
| Blue-collar | 432 (26.3) | 46.3 (41.0-51.6) | 0.56***(0.42-0.75) | |
| Not working[a] | 609 (40.6) | 60.4 (56.2-60.2) | Ref. | |
| Marital status | | | | |
| Single | 378 (22.6) | 39.5 (28.9-51.2) | Ref. | Ref. |
| Married | 1107 (68.7) | 59.5 (55.5-63.4) | 2.24**(1.33-3.77) | 1.44(0.79-2.61) |
| Widowed/ divorced | 117 (8.7) | 74.8 (63.8-83.3) | 4.53***(2.17-9.43) | 1.95(0.88-4.32) |
| Have a child | | | | |
| Yes | 1169 (74.4) | 62.1 (29.3-50.6) | 1.68*(1.08-2.60) | |
| No | 433 (25.6) | 39.5 (58.0-66.0) | Ref. | |
| Wealth status | | | | |
| Q1 (poorest) | 217 (20.0) | 66.6 (60.4-72.2) | Ref. | Ref. |
| Q2 | 301 (20.3) | 53.6 (44.7-62.2) | 0.58*(0.38-0.88) | 0.64*(0.44-0.94) |
| Q3 | 333 (20.4) | 52.7 (44.7-60.5) | 0.56**(0.37-0.84) | 0.54**(0.36-0.81) |
| Q4 | 381 (19.3) | 53.0 (46.8-59.0) | 0.56**(0.40-0.79) | 0.64*(0.45-0.91) |
| Q5 (richest) | 370 (20.0) | 55.7 (48.6-62.5) | 0.63*(0.43-0.92) | 0.71(0.46-1.08) |
| Religion | | | | |
| Muslim | 1252 (92.3) | 56.0 (51.8-60.1) | 0.84(0.60-1.18) | 0.67(0.35-1.27) |
| Others[b] | 350 (7.8) | 60.1 (52.3-67.4) | Ref. | Ref. |
| Place of residence | | | | |
| Urban | 968 (66.8) | 59.6 (54.1-64.9) | 1.50**(1.12-2.01) | 1.50*(1.02-2.21) |
| Rural | 634 (33.2) | 49.6 (45.0-54.2) | Ref. | Ref. |
| Province | | | | |
| West Sumatera | 255 (5.2) | 53.0 (47.2-58.8) | 0.75(0.52-1.08) | 0.93(0.56-1.54) |
| DKI Jakarta | 264 (9.5) | 59.9 (53.3-66.1) | Ref. | Ref. |
| West Java | 275 (42.1) | 54.4 (46.7-61.8) | 0.80(0.53-1.20) | 0.89(0.54-1.45) |
| East Java | 264 (31.2) | 58.7 (51.5-65.5) | 0.95(0.64-1.41) | 1.29(0.79-2.10) |
| Balinese | 279 (4.4) | 54.9 (49.7-60.0) | 0.81(0.58-1.14) | 0.57(0.27-1.22) |
| South Sulawesi | 265 (7.6) | 55.6 (48.2-62.8) | 0.84(0.56-1.25) | 0.97(0.58-1.64) |

OAO, overweight and obesity; AOR, adjusted odd ratio (adjusted for all variables in the model); CI, confidence interval; COR, crude odd ratio. Ref. Reference group.

[a]Housewives, student, and not working

[b]Christian, Catholic, Hinduism, Buddhist, Confucianism

*** p < 0.001; ** p < 0.01; * p < 0.05

**Table 2. Wagstaff normalised concentration index of overweight and obesity by wealth status and education level.**

| Rank variable, sample | n | WCI | Robust SE | p-value | 95% CI |
|---|---|---|---|---|---|
| **Wealth status** | | | | | |
| Overall | 1602 | −0.073 | 0.028 | 0.012 | −0.129 to −0.017 |
| Males | 844 | −0.105 | 0.052 | 0.048 | −0.207 to −0.003 |
| Females | 758 | −0.067 | 0.043 | 0.124 | −0.152 to 0.017 |
| **Education level** | | | | | |
| Overall | 1602 | 0.062 | 0.028 | 0.029 | 0.008 to 0.117 |
| Males | 884 | 0.224 | 0.044 | <0.0001 | 0.129 to 0.306 |
| Females | 758 | −0.095 | 0.070 | 0.179 | −0.231 to 0.031 |

CI, confidence interval; SE, standard error; WCI, Wagstaff normalised concentration index.

smaller. That is, while OAO were initially more prevalent among wealthier Indonesians, the trend since 2000 indicates that OAO is increasingly affecting poorer segments of the population [15]. This sign of nutrition transition is then strengthened by our finding using data collected in 2018: OAO were more concentrated among low-income populations. The shift in the global food system makes unhealthy foods more accessible and cheaper—along with exploitative marketing practices and packaging—directly linked to the growing OAO prevalence. This may have more effect among the low-income group as they have limited access to nutritious foods, such as fruits and vegetables, as well as access to weight-management activities [4].

OAO were more concentrated among more educated participants. This was in line with previous analysis on data from 76 cities in 8 Latin American countries [25]. In addition, a national survey conducted in 2018 also reports that the prevalence is higher among more educated adults in Indonesia [26]. More educated individuals have the skills and knowledge for outsourcing housework and childcare that result in less moderate physical activity spent [27], which leads to a higher likelihood of being overweight/ obese. As shown by our subgroup analysis, this education-related inequality was only significant among men. Our subgroup analysis showing that wealth- and education-related inequality were only significant among males indicated that the pattern of OAO inequality differs by gender. In other words, there was a sign of gender as an effect modifier, as highlighted in a previous review [28].

Our findings revealed that significant wealth- and education-related socioeconomic inequalities in OAO were observed exclusively among males. This gender-specific pattern suggests that sociocultural and economic factors distinctly influence OAO risk by sex in Indonesia. Traditional gender roles often assign women greater household and caregiving responsibilities, limiting their exposure to obesogenic environments characterised by sedentary occupations and urban lifestyles more prevalent among men [29]. Men, particularly in urban and formal employment sectors, may have greater access to high-calorie foods and reduced physical activity, which could amplify wealth- and education-related disparities in OAO risk [30]. These observations align with previous research highlighting gender as an effect modifier in the socioeconomic gradient of obesity [16].

Although OAO were generally more concentrated among lower SES by wealth and higher SES by education, there were variations according to several characteristics, mainly by gender, age, marital status, wealth status, and place of residence. All of these variables were also the determinants of OAO according to our regression model, except for marital status which was fitted as a confounding variable. Results from the inequality analysis indicated that most of these variables were the main contributors for the inequality in OAO. The subsequent paragraphs discussed the role of these variables.

Education as a covariate contributed to the wealth- and education-related inequality in OAO. Within each group of population with higher education (secondary and higher education), OAO were more concentrated among the poor indicated

## Table 3. Decomposition of inequalities in overweight and obesity by wealth status and education level among Indonesian adults.

| Variables | Elasticity | Wealth status | | | Education level | | |
|---|---|---|---|---|---|---|---|
| | | WCI | Pct,% | Summed pct., % | WCI | Pct,% | Summed pct., % |
| Sex, ref. male | −0.477 | | | −20.4 | | | −32.1 |
| Female | 0.423 | 0.035 | −20.4 | | −0.045 | −32.1 | |
| Age, ref. 18–25 years | | | | 1.8 | | | −68.6 |
| 26–35 years | 0.124 | −0.102 | 17.4 | | 0.114 | 23.6 | |
| 36–45 years | 0.145 | 0.078 | −15.6 | | −0.024 | −5.9 | |
| ≥46 years | 0.241 | 0.000 | 0.0 | | −0.215 | −86.3 | |
| Education level, ref. primary or lower | | | | 29.7 | | | 183.1 |
| Secondary | 0.152 | −0.104 | 21.6 | | 0.593 | 149.9 | |
| College/ university | 0.050 | −0.117 | 8.1 | | 0.397 | 33.2 | |
| Occupation, ref. not working[a] | | | | 9.6 | | | 17.9 |
| Professional | −0.004 | 0.158 | 0.9 | | 0.071 | −0.5 | |
| White-collar | 0.068 | −0.075 | 7.0 | | 0.123 | 14.0 | |
| Blue-collar | −0.010 | 0.119 | 1.7 | | −0.254 | 4.4 | |
| Marital status, ref. single | | | | −3.0 | | | −45.1 |
| Married | 0.204 | 0.023 | −6.4 | | −0.111 | −37.9 | |
| Widowed/divorced | 0.047 | −0.053 | 3.4 | | −0.093 | −7.3 | |
| Religion, ref. others[b] | | | | 6.1 | | | 42.5 |
| Muslim | −0.350 | 0.013 | 6.1 | | −0.073 | 42.5 | |
| Wealth status, ref. Q1 (poorest) | | | | 79.6 | | | 7.1 |
| Q2 | −0.077 | −0.498 | −52.7 | | 0.016 | −2.1 | |
| Q3 | −0.108 | 0.013 | 1.9 | | 0.024 | −4.4 | |
| Q4 | −0.070 | 0.505 | 48.7 | | −0.049 | 5.8 | |
| Q5 (richest) | −0.060 | 1.000 | 81.7 | | −0.078 | 7.7 | |
| Place of residence, ref. rural | −0.105 | | | 81.4 | | | 99.4 |
| Urban | 0.213 | −0.279 | 81.4 | | 0.280 | 99.4 | |
| Province, ref. DKI Jakarta | | | | −18.7 | | | −12.3 |
| West Sumatera | −0.004 | 0.199 | 1.1 | | −0.016 | 0.1 | |
| West Java | −0.037 | −0.239 | −12.2 | | 0.001 | −0.1 | |
| East Java | 0.073 | 0.101 | −10.0 | | −0.076 | −9.2 | |
| Balinese | −0.023 | 0.080 | 2.5 | | 0.084 | −3.2 | |
| South Sulawesi | −0.001 | 0.047 | 0.1 | | −0.022 | 0.0 | |
| *Total explained* | | | 166.3 | | | 185.8 | |
| *Residual* | | | −66.3 | | | −85.8 | |

Pct., percent of contribution; Ref. reference group; WCI, Wagstaff normalised concentration index.

[a]Housewives, student, and not working

[b]Christian, Catholic, Hinduism, Buddhist, Confucianism

by the negative WCIs. This result was consistent with previous research [16,31]. Regarding wealth status as covariate, we found it as an important contributor for the inequality by wealth status, and this was consistently shown in previous studies [16,32]. However, its total contribution to education-related inequality was relatively small due to the different direction of the WCI within categories of wealth status. The WCI of the poorer groups—Q2 and Q3—was negative, indicating that OAO was more concentrated among individuals with lower education. However, the WCIs for the richer groups were positive, suggesting that OAO was more prevalent among those with higher education.

The findings related to wealth status and education indicated that the effect of wealth status on OAO varied across education levels. In other words, the effect of wealth status on OAO was stronger than that of education, which was also found in a study among adults in Saudi Arabia [33]. Education has a less direct impact on obesity since it takes systemic assistance (such as access to healthy food options) to turn knowledge into action. In addition to financial constraints, cultural issues may influence the impact of education on OAO. Therefore, to effectively combat obesity, measures that target wealth-driven behaviors—like taxing unhealthy foods or expanding access to reasonably priced, nutrient-dense options for all socioeconomic groups—must be combined with educational initiatives.

Consistent with other studies [15,32], our results showed that place of residence was among the primary contributors of wealth- and education-related inequality in OAO. Urban populations were more likely to be living with OAO, especially among the poorer groups, as indicated by the positive elasticity and the negative WCI showing a concentration of OAO in urban residents mostly coming from lower wealth quintiles. This pattern contributed substantially to wealth-related inequality. The higher likelihood of OAO among urban population is attributed mainly to the obesogenic environment prevalent in urban areas. This environment facilitates weight gain and one that hinders weight loss, as it promotes the availability and accessibility of cheap food rich in fat and sugar, as well as a high dependence on car use, discouraging walking, bicycling to schools/workplaces, or outdoor physical activities during leisure. Its social neighborhoods are relatively more unsafe and stressful [34]. This type of environment is a result of modernisation processes increasing the risk of OAO in developing countries, including Indonesia. Previous research shows that people in the lower socioeconomic groups (low income, low education, live in poor and unsafe neighborhoods) are at higher risk [34], which was in line with our findings on wealth status and OAO. Interestingly, this study also found that the prevalence of urban participants living with OAO was higher among more educated groups. Living in an urban area with better education creates a greater opportunity to get employed in less physically demanding occupation. Also, they may be less involved in domestic chores as they have the ability to delegate this task and to allow them to have more leisure time. These findings highlighted that urban populations with lower income were the most disadvantaged group, while the effect of education could not mitigate the risk of OAO. To address these disparities, it is essential to implement targeted interventions that enhance access to nutritious foods and opportunities for physical activity, while also improving health literacy in both urban and rural environments.

Results showed that females were more likely to be living with OAO, and this group was more concentrated among more affluent populations, consistent with findings from previous research [32]. Females have a higher probability of living with OAO than their counterpart due to a combination of biological, hormonal, and sociocultural factors [35]. This risk becomes higher if they come from wealthier families as they can afford more calories in their diets. In addition, they may be less likely to be physically active due to less engagement in domestic work and more likely to be employed in physically demanding jobs. Moreover, societal gender norms restrict women's public physical activity, thereby limiting exercise opportunities and contributing to elevated obesity rates [6]. Women are more likely than men to face situations, such as family or social gatherings, where rejecting food offered by family or friends is perceived as disrespectful. This dynamic complicates their ability to manage portion sizes or dietary choices without causing offence to others [36]. Social events often centre on calorie-dense traditional foods, which complicates the maintenance of healthy weights [37]. Additionally, women from wealthier families may attend such events more frequently than those from less affluent backgrounds; consequently, OAO among females were more concentrated within the wealthier groups. However, result of the inequality analysis by education level indicated that OAO among females were more concentrated among those with lower educational attainment. The findings indicate that education may mitigate the obesity-promoting effects of wealth in women. It is essential to enhance OAO interventions, including nutrition education and physical activity promotion, that are attuned to cultural and social norms.

Our results support previous analysis [31] that OAO were more prevalent among older groups, as indicated by the increasingly positive elasticities among the older groups as well as the AOR. Age is positively associated with OAO due to a combination of physiological, metabolic, and lifestyle changes that occur as people grow older, which creates a

predisposition to weight gain and fat accumulation over time [38]. Older people often engage in less physical activities due to retirement, health conditions, or reduced mobility, which then contribute to a positive energy balance and promote weight gain over time [38]. As the concentration of OAO across categories of wealth status varied, the total contribution of age to the wealth-related inequality was relatively smaller, which supports a previous finding [32]. On the other hand, this factor reduced the observed education-related inequality due to the positive elasticities in all age groups and the negative WCI, especially among the oldest age group.

The decomposition analysis further showed that age and marital status had substantial negative contributions to education-related inequalities. The negative contribution of age likely reflects that older adults with lower education levels bear a disproportionate burden of OAO, possibly due to cumulative adverse lifestyle and metabolic exposures over the life course [39]. Concerning marital status, its negative contribution may arise because married and widowed individuals exhibit higher OAO prevalence irrespective of education, thereby offsetting positive inequality contributions from other variables. Sociocultural norms in Indonesia, such as interpretations of obesity in married women as a sign of success [40], may further complicate these dynamics. These findings highlight the complex interplay of sociodemographic factors shaping socioeconomic disparities in OAO and emphasize the necessity of interventions sensitive to gender, age, and marital context within Indonesia's nutrition transition.

Further, our results showed that OAO were more prevalent among more educated participants in the age of 26–35 years, while for the older age groups, individuals with lower education levels were more prone to OAO. This finding was in line with previous research on the Korean population [41], where the probability of OAO across education levels varied by age. By wealth status, OAO were more concentrated among the lower SES for those aged below 36 years but for the older individuals, OAO were more common among the wealthier groups. The results demonstrate that the impact of wealth and education differs among age groups. Consequently, weight-loss management programs and obesity interventions must be customised to meet individual needs, ensuring appropriate calorie intake and safe physical activity, particularly for older adults.

In a previous study, marital status has a positive contribution to wealth-related inequality in OAO [16], while according to our analysis, this factor had no substantial contribution. However, we did find that the values of elasticities for marital status in both studies were similarly positive indicating the married and divorced/widowed people were more likely to be living with OAO than unmarried individuals, which supported the result of a study in China [42]. Single or unmarried may have a higher concern on body image to attract potential partners, while married individuals may experience changes in their lifestyle habits due to the changes in family structure and responsibilities [42]. Another explanation, particularly for unmarried women, comes from a qualitative study in Indonesia revealing that the society views obesity in married women as a sign of success, but single obese women suffer psychological issues due to acquaintances mocking them about their physical size [40]. Further, our results suggested that marital status reduced the education-related inequality in OAO. As subpopulations with the higher likelihood of overweight and obesity (positive elasticity) by marital status were more concentrated among less educated groups (negative WCI), this factor recorded a negative contribution to the observed inequality. The disproportionate distribution of OAO by marital status and education levels are also captured in previous research [43].

Socioeconomic status (SES) is widely acknowledged as the key determinant of various health outcomes including OAO. Our analysis provided evidence of the observed inequalities by two measures of SES—wealth status and education level—which varied across several variables. These observed inequalities in OAO along with its growing prevalence could challenge the government's ambitious goal to reduce obesity prevalence to below 3% by 2030, especially given the structural barriers faced by the poor. In addition, populations in the lower socioeconomic groups according to the wealth status were more likely to be living with OAO according to our finding indicating the magnitude of the problem. Therefore, it is crucial to address the barriers particularly for the poor in accessing programs related to OAO.

Programs or policies in the food system, such as mandatory nutrition labelling, tax on sugar-sweetened beverages or ultra-processed snacks, and their advertising restrictions could be considered to discourage the consumption of these unhealthy foods, especially in urban settings. At the same time, the country's current policy on subsidizing rice, known as *Raskin* (formerly) or *Rastra* [44], without parallel attention to improve fruits, vegetables, or lean protein consumption, needs further careful considerations. A diet predominantly composed of rice may lack the variety of nutrients essential for optimal health. Reallocating subsidies to encompass a broader array of healthy foods may enhance overall nutritional outcomes and subsequently decrease the prevalence of OAO. Furthermore, given that OAO are more prevalent among women, enhancing women's economic empowerment may alleviate the burden of OAO by facilitating their participation in income-generating activities, which can lead to improved dietary health, household income, and physical activity [45]. Additionally, it is essential to address cultural norms and beliefs associated with OAO, as research indicates that these factors not only affect individual health behaviours but also sustain systemic disparities in obesity prevalence [36,37,40].

This study comes with a number of limitations. First, results on the observed inequalities and their contributing factors should be seen as association rather than causality since the data were collected at one point in time. Further research with a longitudinal design is encouraged to look at the shifts of inequalities in OAO among Indonesian adults. Previous studies have indicated these changes, but further exploration using updated data is needed to capture current phenomena. Secondly, future research should capture other variables that could be relevant contributors to OAO, such as physical activity, food habits and environment, as we did not include them in the current analysis. Despite the limitations, our study is of importance as we analyzed the inequalities in OAO using two measures of socioeconomic factors, in which, to the best of our knowledge, similar analysis for the Indonesian population has not been conducted. Information on OAO was obtained from anthropometric measures that were taken by trained enumerators at the time of data collection, which increased the accuracy of the data.

## Conclusion

To conclude, the results indicated a pro-poor inequality in OAO, while by education level, the outcome was more concentrated among more educated participants. OAO was more prevalent among females but the disproportionate distribution across wealth and education levels only occurred among males. Place of residence and education were the primary contributors of the inequalities, while the contribution of wealth status was more noticeable in the inequality by wealth status. The existing socioeconomic inequalities may increase the burden of NCDs and healthcare costs, exacerbating health disparities and ultimately reducing the quality of life for vulnerable populations. Strategies to reduce inequalities are urgently needed, equally as important as those designed to control and prevent the burden of OAO. These strategies may include policies and interventions within the food system, such as nutrition labelling, taxation on unhealthy foods, and subsidies for healthy foods, alongside nutrition education that considers cultural norms and beliefs associated with OAO.

## Supporting information

**S1 Table. Sample characteristics and data completeness.** The table describes the completeness of data and sample exclusions in the Indonesian Food Barometer dataset.
(DOCX)

**S2 STROBE Checklist. Completed STROBE checklist for observational studies.** The checklist indicates where each item of the STROBE guideline is reported within the manuscript.
(PDF)

## Author contributions

**Conceptualization:** Siti Nurokhmah, Judhiastuty Februhartanty, Helda Khusun.

**Data curation:** Siti Nurokhmah, Judhiastuty Februhartanty.

**Formal analysis:** Siti Nurokhmah.

**Supervision:** Judhiastuty Februhartanty, Helda Khusun.

**Validation:** Siti Nurokhmah, Judhiastuty Februhartanty, Helda Khusun.

**Visualization:** Siti Nurokhmah.

**Writing – original draft:** Siti Nurokhmah, Judhiastuty Februhartanty, Helda Khusun.

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
