## [Decision Letter · Decision Letter 0]

13 Feb 2025

Dear Dr. Nurokhmah,

Please provide responses to the reviewers below.

Please submit your revised manuscript by Mar 30 2025 11:59PM If you will need more time than this to complete your revisions, please reply to this message or contact the journal office at plosone@plos.org . A rebuttal letter that responds to each point raised by the academic editor and reviewer(s). You should upload this letter as a separate file labeled 'Response to Reviewers'.A marked-up copy of your manuscript that highlights changes made to the original version. You should upload this as a separate file labeled 'Revised Manuscript with Track Changes'.An unmarked version of your revised paper without tracked changes. You should upload this as a separate file labeled 'Manuscript'.

We look forward to receiving your revised manuscript.

Kind regards,

Fitriana Murriya Ekawati

Academic Editor

PLOS ONE

Journal Requirements:

2. In the online submission form, you indicated that [Public availability of the dataset from the 2018 Indonesian Food Barometer by SEAMEO RECFON based on Open Science philosophy is in progress. Raw data sets are available on request from the corresponding author.].

Reviewers' comments:

Reviewer's Responses to Questions

**Comments to the Author**

1. Is the manuscript technically sound, and do the data support the conclusions?

Reviewer #1: Yes

Reviewer #2: Partly

Reviewer #3: Yes

2. Has the statistical analysis been performed appropriately and rigorously?

Reviewer #1: I Don't Know

Reviewer #2: No

Reviewer #3: Yes

3. Have the authors made all data underlying the findings in their manuscript fully available?

Reviewer #1: Yes

Reviewer #2: No

Reviewer #3: Yes

4. Is the manuscript presented in an intelligible fashion and written in standard English?

Reviewer #1: Yes

Reviewer #2: Yes

Reviewer #3: Yes

Reviewer #1: Manuscript Title: Socioeconomic Inequalities in Overweight and Obesity Among Adults: Results from the Indonesian Food Barometer Study

The manuscript presents an important and timely study investigating the socioeconomic inequalities in overweight and obesity among Indonesian adults, based on data from the Indonesian Food Barometer. As the prevalence of overweight and obesity in Indonesia continues to rise, particularly across different socioeconomic strata, this study provides valuable insights into the factors contributing to this trend. The authors present a comprehensive analysis of how wealth status, education levels, and place of residence contribute to the observed inequalities in overweight and obesity in Indonesia. Given the global importance of addressing the dual burden of malnutrition, this research makes a significant contribution to the existing literature.

Specific Suggestions:

Introduction:

The introduction is well-written and clearly sets the stage for the study. However, I recommend further elaborating on the concept of the "nutrition transition" in Indonesia. While the manuscript touches on this, a bit more background on how this transition has influenced both undernutrition and overnutrition could provide greater context for the study's findings.

It would be helpful to briefly mention the existing literature on the relationship between socioeconomic status and obesity in other countries, as this would position the Indonesian findings in a broader international context.

Methods:

The methods section is generally clear, but it would be useful to provide a bit more detail about how the socioeconomic status (SES) was measured. Was it based on income, wealth quintiles, or a combination of factors? A brief description of how SES was categorized would clarify the analysis for readers.

It would also be useful to include more details about how education levels were categorized. Were they grouped by specific attainment levels (e.g., primary, secondary, tertiary), and how were these related to the socioeconomic status stratifications?

Results:

The results are well-presented, but there is room for more in-depth analysis of the relationship between place of residence and obesity levels. Are there significant regional disparities in the prevalence of overweight and obesity, particularly between urban and rural areas? Further exploration of how geographic location might influence dietary patterns, access to healthcare, and lifestyle factors would be valuable.

Additionally, it may be helpful to include some basic demographic information on the sample population, such as age distribution and gender, to allow readers to better understand the characteristics of the sample.

Discussion:

The discussion provides a solid interpretation of the results and contextualizes the findings within the broader literature. However, the authors should consider expanding on the policy implications of the study. Given the increasing prevalence of obesity in Indonesia, what public health interventions might help address the socioeconomic disparities observed in this study? Recommendations for future research could also be included, particularly regarding potential interventions targeting at-risk populations.

While the study highlights the role of education levels and wealth status in driving inequalities, it would be interesting to see more discussion on the role of cultural factors, urbanization, and food environments in shaping obesity trends across different socioeconomic groups.

Conclusion:

The conclusion is concise and effectively summarizes the main findings. However, it would be useful to more explicitly state the potential consequences of these inequalities for Indonesia's public health system and what specific actions might be needed to address these growing disparities in obesity prevalence.

Figures and Tables:

The manuscript would benefit from a clearer presentation of the data in the tables and figures. Specifically, the tables could be more informative if they included effect sizes or confidence intervals for the reported associations, which would give readers a better sense of the magnitude of the relationships observed.

The figures should also be more clearly labeled, with additional information in the legends to help readers fully understand the data being presented.

Minor Technical Corrections:

In the abstract, the sentence structure could be improved for clarity. For example, instead of saying "The data from the Indonesian Food Barometer represent around a half of Indonesian population," it could be rephrased as "The data from the Indonesian Food Barometer represent approximately half of the Indonesian population."

There are a few instances where the authors use "obesity" and "overweight" without defining the specific cutoffs used to classify individuals in these categories. It would be helpful to include these definitions for the sake of transparency.

Reviewer #2: Dear authors,

Thank you for your submission. This manuscript, investigating socioeconomic inequalities in overweight and obesity (OAO) among Indonesian adults, addresses a critical public health issue with the potential for significant impact. However, the current presentation and methodology require substantial revisions to meet the standards for publication in PLOS ONE. While the research topic is highly relevant, several methodological, analytical, and interpretive limitations weaken the study's overall impact.

My primary concern revolves around the methodological approach. While the chosen statistical method, the Wagstaff normalized concentration index (WCI), is relevant for analysing socioeconomic inequalities, the manuscript lacks a robust justification for its selection. The authors do not fully explain why the WCI was chosen over other possible methods, and also does not sufficiently discuss the rationale behind using the 'svy' command in Stata, which could influence the robustness of the results. It is vital to provide a more detailed explanation of the methodology to convince the scientific community of the findings. The cross-sectional study design is also a major concern. While the authors have collected relevant data, making causal statements from cross-sectional data weakens the validity of the conclusions, particularly when discussing the complex interplay between socioeconomic factors and health outcomes. Therefore, it is important for the authors to acknowledge the study's limitations and ensure that the inferences made from the data are accurate.

Furthermore, the discussion section requires significant strengthening. Some conclusions are made without adequate support from the presented data, and the interpretations are not consistently connected with existing literature, particularly in areas related to physical activity. The authors must focus the discussion on presenting clear justifications for all the claims they make based on the statistical findings. The absence of a clear link with studies on physical activity, especially concerning older adults, limits the broader relevance of the findings, especially given the target population of the study. It is, therefore, important to contextualize the data by including studies on physical activity and cognitive impairment in the geriatric population. Furthermore, the discussion does not provide concrete, actionable policy implications for this area, so the authors need to focus more on creating recommendations for the target population.

The manuscript also fails to meet PLOS ONE's data availability criteria. All the data should be provided as part of the manuscript or its supporting information, or deposited to a public repository, not just submitting the analysed and summarised ones. Furthermore, the reference list requires significant revision to ensure all sources are updated and from within the last 10 years. Proper and appropriate citations are essential for the integrity of scientific publications.

While using an appropriate index, the statistical analysis needs to be applied more rigorously. This is particularly evident in the cross-sectional limitations and the need for detailed descriptions of the analysis. The conclusions, at present, are only partially supported by the data. Therefore, this manuscript needs a major revision. The authors should re-evaluate their methodology to provide a more robust justification for the statistical choices and address all limitations of cross-sectional data while providing well-detailed data analysis and interpretations. In addition, the discussion needs strengthening by linking it to existing literature, updating all the references, providing full data access, and providing actionable recommendations for future research and policy. Failure to address these significant concerns will prevent the manuscript from being considered for publication.

Reviewer #3: The topic is highly relevant and the research was very well conducted, but it is not new. It would be interesting if the authors could present suggestions for effective changes by governments and populations in order to reduce inequality and the prevalence of overweight and obesity. It is not enough to say that strategies are necessary; proposals must be presented in one or two paragraphs of the discussion, so that measures can be taken to find solutions.

**Do you want your identity to be public for this peer review?** For information about this choice, including consent withdrawal, please see our Privacy Policy

Reviewer #1: No

Reviewer #2: **Yes: ** Alston Choong

Reviewer #3: **Yes: ** Roberto Fernandes da Costa

---

## [Author Response · Author response to Decision Letter 1]

22 Mar 2025

Response to the reviewers' comments can be found in the attached file "Response to Reviewers" along with this submission

---

## [Decision Letter · Decision Letter 1]

24 Jun 2025

Dear Dr. Nurokhmah,

Thank you for submitting your manuscript to PLOS ONE. After careful consideration, we feel that it has merit but does not fully meet PLOS ONE’s publication criteria as it currently stands. Therefore, we invite you to submit a revised version of the manuscript that addresses the points raised during the review process.

We look forward to receiving your revised manuscript.

Kind regards,

Fitriana Murriya Ekawati

Academic Editor

PLOS ONE

Journal Requirements:

Reviewers' comments:

Reviewer's Responses to Questions

**Comments to the Author**

Reviewer #2: (No Response)

Reviewer #3: All comments have been addressed

2. Is the manuscript technically sound, and do the data support the conclusions?

Reviewer #2: Yes

Reviewer #3: Yes

3. Has the statistical analysis been performed appropriately and rigorously?

Reviewer #2: Yes

Reviewer #3: Yes

4. Have the authors made all data underlying the findings in their manuscript fully available?

Reviewer #2: No

Reviewer #3: Yes

5. Is the manuscript presented in an intelligible fashion and written in standard English?

Reviewer #2: Yes

Reviewer #3: Yes

Reviewer #2: Dear Authors,

Thank you for submitting your revised manuscript. This study addresses a highly relevant public health issue in Indonesia, examining the complex interplay between socioeconomic status (wealth and education) and overweight/obesity (OAO). The application of the Wagstaff normalized concentration index and decomposition analysis provides valuable quantitative insights into the nature and drivers of these inequalities, contributing significantly to the literature in this area, particularly within the Indonesian context. The manuscript is generally well-written and methodologically sound.

While the core analysis is strong, a few points require clarification and refinement to ensure the manuscript meets its full potential and adheres strictly to PLOS ONE policies:

1. Crucial Clarification of Data Collection Period: There remains significant ambiguity in the Methods section (Lines 77-78) regarding the data collection dates. The text states, "The 2018 Indonesian Food Barometer (IFB) survey collected data from March 2009 to January 2010". This wording is confusing and requires immediate correction for clarity. Please explicitly clarify whether the analysis uses data from a survey fielded in 2018, or if it uses data collected during 2009-2010 as part of a survey labeled or reported in 2018. If the data are indeed from 2009-2010, the manuscript framing needs adjustment, and the timeliness/relevance requires stronger justification given the rapid nutrition transition since that period.

2.Data Availability Statement: Your manuscript states data access is restricted due to institutional/governmental policy and available upon request from SEAMEO RECFON. PLOS ONE's policy requires unrestricted access unless an exception is approved. "Available upon request" is generally insufficient. Please work directly with the PLOS ONE editorial office to validate this exception. This requires providing verifiable details of the restricting policy and ensuring the SEAMEO RECFON access route is confirmed as compliant with the journal's exception standards. This policy point must be formally resolved for the manuscript to proceed.

3. Role of Initial Logistic Regression: The manuscript briefly mentions performing logistic regression "before undertaking the inequality analysis" (Line 117). The precise purpose and how the results of this initial analysis informed the primary inequality analysis remain slightly unclear. Please briefly clarify this connection or rephrase to avoid potential reader confusion about the role of this preliminary step.

4. Discussion Nuance: The discussion is solid but could benefit from slightly deeper interpretation in places. For example, the finding that significant wealth and education inequalities were observed only among males is interesting. Could you offer slightly more developed speculation, citing relevant literature on potential contributing factors specific to Indonesia (e.g., gender roles, labor market participation)? A more detailed exploration of why certain factors (like age and marital status) had large negative contributions in the decomposition analysis could also add valuable nuance.

5. Reference Recency: While generally good, please perform a final check on references cited for broad background information (e.g., global statistics) to ensure the most current sources are used, especially if older than 10 years. Foundational methodological citations are exempt.

6. Minor Wording Consistency: Please ensure consistent terminology throughout, for example, using "had OAO" consistently instead of variations like "lived with OAO".

In conclusion, this is a methodologically sound and important study. Addressing the points above, particularly the critical clarification of the data collection period and ensuring full compliance with the journal's data availability policy (including verification of any necessary exceptions), will make the manuscript suitable for publication in PLOS ONE. These revisions should be straightforward.

Reviewer #3: The authors complied with all my requests, and the modifications resulting from the requests of the other reviewers substantially improved the quality of the manuscript, so I consider that it should be accepted for publication.

**Do you want your identity to be public for this peer review?** For information about this choice, including consent withdrawal, please see our Privacy Policy

Reviewer #2: **Yes: ** Alston Choong

Reviewer #3: **Yes: ** Roberto Fernandes da Costa

---

## [Author Response · Author response to Decision Letter 2]

22 Aug 2025

Summary of Responses to Reviewer and Editor Comments

Reference List: Thoroughly reviewed; no retracted sources. Added four new, current citations (References 28, 29, 38, and 39) as recommended by Reviewer 2.

Data Collection Period: Corrected to “December 2017 to March 2018” (Lines 75–77) for clarity.

Logistic Regression Rationale: Explained that the preliminary logistic regression identified covariates for inclusion in the Wagstaff concentration index decomposition (Lines 116–118).

Data Availability Statement: Expanded to note restricted access via SEAMEO RECFON and inclusion of summary data in Supplementary Table S1; confirmed compliance with PLOS ONE policy.

Discussion Enhancements: Provided deeper interpretation of male-only socioeconomic inequalities and negative contributions of age and marital status, supported by relevant literature (Lines 230–239; 323–333).

Terminology & Reference Recency: Standardized people-first language (“people living with overweight and obesity”) and updated background citations to the most recent sources.

Reviewer 3: Noted and appreciated their positive recommendation for acceptance.

Please see the attached detailed point-by-point response document for full responses and manuscript revisions.

---

## [Decision Letter · Decision Letter 2]

6 Nov 2025

Socioeconomic inequalities in overweight and obesity among adults: results from the Indonesian Food Barometer study

PONE-D-24-58609R2

Dear Dr. Siti Nurokhmah,

We’re pleased to inform you that your manuscript has been judged scientifically suitable for publication and will be formally accepted for publication once it meets all outstanding technical requirements.

Kind regards,

Akingbolabo Daniel Ogunlakin, Phd

Academic Editor

PLOS ONE

Additional Editor Comments (optional):

Reviewers' comments:

Reviewer's Responses to Questions

**Comments to the Author**

Reviewer #2: (No Response)

Reviewer #3: All comments have been addressed

2. Is the manuscript technically sound, and do the data support the conclusions?

Reviewer #2: Yes

Reviewer #3: Yes

3. Has the statistical analysis been performed appropriately and rigorously?

Reviewer #2: Yes

Reviewer #3: Yes

4. Have the authors made all data underlying the findings in their manuscript fully available?

Reviewer #2: Yes

Reviewer #3: Yes

5. Is the manuscript presented in an intelligible fashion and written in standard English?

Reviewer #2: Yes

Reviewer #3: Yes

Reviewer #2: Dear Authors,

Thank you for your thorough revisions and for providing a detailed response to the previous reviewers' comments. The manuscript is now substantially improved, particularly regarding the crucial clarification of the data collection period, the enhanced methodological rationale, and the more nuanced discussion of the gender-specific findings. The study is methodologically sound, using appropriate and rigorous techniques (Wagstaff concentration index and decomposition analysis) to address a highly relevant public health issue in Indonesia. The work is very close to being ready for publication. I offer the following minor suggestions to further enhance the manuscript's clarity and impact before publication.

First, the abstract, while now factually correct, is dense with multiple complex findings. To improve readability and immediate impact, I suggest a minor restructuring. Consider stating the overall population findings first (pro-poor by wealth, pro-rich by education), and then use a clear transition to introduce the key finding that these inequalities were statistically significant only among males. This would create a clearer narrative hierarchy for the reader. For instance, a sentence like, "Subgroup analysis revealed that these inequalities were driven exclusively by findings in males, as no significant inequality was observed among females," could provide this clarity.

Second, the discussion has been significantly strengthened. The study's core finding presents a compelling paradox: while overweight and obesity are concentrated among the poor by wealth, they are simultaneously more prevalent among the highly educated, an effect particularly pronounced in males. To maximise the impact of this finding, you might consider explicitly framing it as a "socioeconomic paradox" within the discussion. This terminology would help to crystallise the central theme for the reader and provides a memorable hook that underscores the complexity of the nutrition transition in this context.

Finally, while the "people-first" language is now correctly and respectfully applied throughout, a final proofread for minor wording inconsistencies would add a last layer of polish. For example, ensuring consistent use of phrases like "people living with OAO" versus "participants with OAO" can enhance the manuscript's professional tone.

These are minor suggestions intended to refine what is now a strong, well-supported, and important contribution to the literature. The study is technically sound, and the conclusions are well-supported by the rigorous analysis. I commend the authors on their diligent and successful revision.

Reviewer #3: The authors responded to all my requests and suggestions, so I believe the manuscript can be published in Plos One. Congratulations!

**Do you want your identity to be public for this peer review?** For information about this choice, including consent withdrawal, please see our Privacy Policy

Reviewer #2: **Yes: ** Alston Choong

Reviewer #3: **Yes: ** Roberto Fernandes da Costa

---

## [Editor Report · Acceptance letter]

PONE-D-24-58609R2

PLOS ONE

Dear Dr. Nurokhmah,

I'm pleased to inform you that your manuscript has been deemed suitable for publication in PLOS ONE. Congratulations! Your manuscript is now being handed over to our production team.

Kind regards,

on behalf of

Dr. Akingbolabo Daniel Ogunlakin

Academic Editor

PLOS ONE